# Recent Progress in Gas Sensors Based on P3HT Polymer Field-Effect Transistors

**DOI:** 10.3390/s23198309

**Published:** 2023-10-08

**Authors:** Si Cheng, Yifan Wang, Ruishi Zhang, Hongjiao Wang, Chenfang Sun, Tie Wang

**Affiliations:** Tianjin Key Laboratory of Drug Targeting and Bioimaging, Life and Health Intelligent Research Institute, Tianjin University of Technology, Tianjin 300384, China

**Keywords:** gas sensors, polymer field-effect transistor, organic semiconductors, organic bioelectronics

## Abstract

In recent decades, the rapid development of the global economy has led to a substantial increase in energy consumption, subsequently resulting in the emission of a significant quantity of toxic gases into the environment. So far, gas sensors based on polymer field-effect transistors (PFETs), a highly practical and cost-efficient strategy, have garnered considerable attention, primarily attributed to their inherent advantages of offering a plethora of material choices, robust flexibility, and cost-effectiveness. Notably, the development of functional organic semiconductors (OSCs), such as poly(3-hexylthiophene-2,5-diyl) (P3HT), has been the subject of extensive scholarly investigation in recent years due to its widespread availability and remarkable sensing characteristics. This paper provides an exhaustive overview encompassing the production, functionalization strategies, and practical applications of gas sensors incorporating P3HT as the OSC layer. The exceptional sensing attributes and wide-ranging utility of P3HT position it as a promising candidate for improving PFET-based gas sensors.

## 1. Introduction

With the rapid development of the economy, human quality of life has significantly improved. However, this progress has also led to a series of accompanying environmental issues [1]. In the present era, industrial activities often lead to the emission of non-standard household exhaust gases, consequently releasing a substantial quantity of toxic and harmful substances into the environment. The real-time monitoring of these noxious gases has assumed paramount significance, to ensure that their levels remain within safe thresholds to safeguard human health and the environment [2,3,4]. The advancement of sensors possessing robust stability, heightened sensitivity, and exceptional specificity holds enduring significance for the effective detection of toxic and harmful gases. This progress has far-reaching implications in realms such as food safety, transportation, environmental monitoring, national defense, and various other critical domains [5,6]. Unlike conventional methods such as chemiluminescence, electrochemical, resistance, and optical sensing, PFET-based gas sensors stand out as a promising contender in the realm of gas sensing. This distinction arises from their inherent advantages, including simplicity, user-friendliness, effective analysis of high-throughput signals, and cost-effectiveness [7,8,9].

In PFET-based devices, OSCs serve as carrier transmission channels and sensing elements. They enable the transport of charge carriers through interconnected π bonds between molecules, serving both as current-carrying transmission channels and sensing elements within PFETs [10]. As a result of the aforementioned carrier transport characteristics, specific interactions arise between OSCs and gas molecules. These interactions encompass phenomena such as dipole interactions, van der Waals forces, and others. Consequently, alterations occur in electrical properties, including field-effect mobility and threshold voltage. This orchestrated interplay serves the ultimate goal of enabling the real-time monitoring of external gases [11]. Hence, the selection of the active layer comprising OSCs stands out as a paramount strategy to attain high-performance gas sensors. Crafting functional OSCs with distinctive molecular structures has emerged as a potent approach to enhance the sensitivity, selectivity, and stability of PFET-based gas sensors [12,13,14]. Nonetheless, the synthesis of novel materials often proves intricate and time-intensive. Presently, a comparatively straightforward and efficient approach involves the manipulation of the microstructure of OSCs to generate increased specific surface area, facilitating enhanced gas interactions and reactions [15].

In this comprehensive review, we present an overview of the research advancements pertaining to PFET-based gas sensors utilizing P3HT as a sensing layer across various gas sensing applications. Initially, we provide a succinct overview of the conjugated polymer P3HT. Subsequently, we delve into an exploration of the mechanisms, fabrication techniques, and strategies for functionalization of PFET-based gas sensors incorporating P3HT as the sensing layer. Furthermore, we probe into a discussion of the practical applications of these gas sensors within intricate gas environments, highlighting their effectiveness in efficiently detecting gases such as NH_3_, SO_2_, and NO_x_. Lastly, we undertake a comprehensive analysis of the future prospects and potential applications of these gas sensors. This review serves to underscore the intimate interplay between PFET-based gas sensors featuring P3HT as a sensing layer and the gas environment, significantly propelling the advancement of polymer-based gas sensors.

## 2. Polymer Field-Effect Transistors (PFETs)

Polymer field-effect transistors (PFETs) are a powerful driving force in the modern information technology industry. Since their introduction in the mid-20th century, the inherent advantages of PFETs have been utilized, including miniaturization, ease of preparation, and flexibility in material design and synthesis. These properties pave the way for a variety of applications in clinical disease diagnosis and intelligent health monitoring [16,17].

### 2.1. Classification and Working Mechanism of PFETs

In simple terms, a PFET device is a three-terminal electronic component consisting of four key elements: a gate, an insulating layer, a source–drain electrode, and a OSC layer. Governed by the gate voltage (*V*_G_), charge carriers are introduced from the source electrode, forming an accumulation layer at the interface of several molecular layers thick of the OSC material that comes into contact with the insulating layer. Subsequently, these charge carriers move directionally within the conductive channel, regulated by the source-drain voltage (*V*_SD_), resulting in the generation of a source–drain current (*I*_SD_). Two main configurations arise based on the gate electrode’s positioning, categorizing PFETs into top-gate and bottom-gate structures. These two structures can be differentiated by the locations of the source–drain electrode and the semiconductor layer. In the present context of gas sensing device structures, the bottom-gate top contact configuration is commonly employed. This configuration’s advantage lies in its notably uniform crystal structure within the OSC layer and the interface between the OSC layer and the insulating layer. Due to the larger contact area between the electrode and the semiconductor in this structure, the contact resistance is minimized. As a result, top contact devices typically outperform bottom contact devices [18]. Taking p-type OSC material as an example, Figure 1 illustrates its working mechanism. When a negative voltage is applied between the source and the gate electrodes, positive charge carriers are injected from the source electrode under the control of the gate voltage, forming an accumulation layer at the interface of several molecular layers in contact between the OSCs material and the insulation layer. Subsequently, these positive charge carriers move within the conductive channel, resulting in the formation of source and drain currents, which are regulated by the source and drain voltages.

Given the transport characteristics of such devices, interactions between specific p-type OSCs and gas analytes (such as quenching, doping, and dipole effects) induce alterations in electrical properties such as field-effect mobility and threshold voltage. These changes enable the instantaneous detection and response to external analyte stimulation. Specifically, the response mechanism of OSC gas sensors to oxidizing gas molecules can be understood as the “doping effect.” Oxidizing gas molecules act as electron acceptors, resulting in a similar hole-doping effect when they interact with p-type semiconductors. This interaction provides additional holes for deep trap states or passivates hole traps within the organic semiconductor. As a result, there is an increase in conductivity and a positive drift in the surge voltage. Conversely, when OSC interacts with a reducing gas, the lone pair electrons in the reducing gas engage with OSC, leading to hole reduction when acting on p-type PFETs. Simultaneously, the reducing gas molecules infiltrate the interface to capture hole charges, reducing hole accumulation and transmission. This results in decreased carrier mobility, thus reducing the source–drain current, and causing a negative offset in the surge voltage [19,20].

### 2.2. Conjugated Polymers

To further delve into the performance and functionality of PFETs, German scientist H. Staudinger, in 1920, posited that polymers are composed of small molecules, known as “monomers”, which undergo polymerization reactions through covalent bonding. The resulting polymer has a relative molecular mass generally exceeding 10,000. For an extended period, polymer materials were largely considered to be inherently electrically insulating. However, in 1967, Hideki Shirakawa broke new ground by fabricating the first polyacetylene film and subsequently enhancing the film’s electrical conductivity through a process called doping. This marked a turning point, challenging the conventional notion of polymers as insulators. In the 1970s, Hideki Shirakawa, alongside Alan J. Heeger and others, managed to amplify the conductivity of polyacetylene films by several orders of magnitude using doping agents such as I_2_ and AsF_5_. This pivotal breakthrough laid the foundation for the exploration and application of conjugated polymers [21,22,23].

In contrast to other inorganic materials, conjugated polymers offer the advantage of easy customization to meet specific requirements through the incorporation of functional groups or the manipulation of physical conditions. Moreover, their capacity for solution-based processing at low temperatures holds the promise of significantly reducing the cost of gas sensor devices. The array of choices for conjugated backbones and side chains in polymers provides ample room for adaptation to various application scenarios. Additionally, conjugated polymer films possess flexibility and lightweight attributes, further enhancing their potential for gas-sensing applications. The multifunctionality exhibited by polymer materials opens up an entirely new era for the advancement of gas detection electronic devices, having the potential to bring about revolutionary changes in society.

The rapid progression in synthetic techniques in recent years have facilitated the preparation of numerous conjugated polymer materials possessing exceptional properties. Common organic polymer materials employed for crafting gas sensors in PFETs encompass P3HT, poly{2,5-bis(2-octyldodecyl)-2,3,5,6-tetrahydro-3,6-dioxopyrrolo[3,4-c]pyrrole-1,4-diyl-alt-[2,2′-(2,5-thiophene)bisthiophene(3,2-b)thiophene]-5,5′-diyl} (PDBT-co-TT), poly(2,5-bis(3-hexadecylthiophen-2-yl)thieno[3,2-b]thiophene) (PBTTT) [24], poly{[N,N-9-bis(2-octyldodecyl)-naphthalene-1,4,5,8-bis(dicarboximide)-2,6-diyl]-alt-5,59-(2,29-bithiophene)} (P(NDI2OD-T2)) [25], and poly(quaternary thiophenes) (PQTs) [26], as illustrated in Figure 2.

For instance, one of the most widely used polymers, P3HT, has garnered substantial attention due to its affordability, ease of processing, high carrier mobility, and potential applications in gas detection, solar cells, and optical transduction. Its characteristics, such as crystallization and orientation, play a pivotal role in shaping its properties. P3HT stands as a semi-crystalline polymer, forming both crystalline and amorphous regions during crystallization. Carrier movement predominantly occurs between these crystalline regions; however, the presence of grain boundaries tends to diminish the carrier mobility of P3HT films. Notably, P3HT molecular chains can adopt three arrangements: edge-on along the alkane chain direction, face-on along the π–π stacking directions, and end-on along the main chain direction, as depicted in Figure 3. Carrier transport through these structures exhibits anisotropic behavior: it is most rapid along the main chain, followed by π–π stacking, and least prevalent along the alkane chain direction. Diverse applications necessitate specific molecular chain stacking. For example, in the context of FETs, the edge-on structure proves more advantageous as it aligns the charge transport direction parallel to the substrate. Conversely, in solar cells, where charge transport should be perpendicular to the substrate, face-on and end-on structures are preferable. Thus, the crystallization and orientation of P3HT can be finely adjusted in line with application prerequisites, thereby enhancing the resulting performance [27,28,29,30].

### 2.3. Crystallisation and Orientation Regulation

The modulation of P3HT crystallization and orientation generally includes enhancing P3HT’s crystallinity, inducing the alignment of P3HT molecular chains (resulting in face-on and end-on structures), and prompting the creation of nanowires or nanopores within P3HT. This review primarily delves into the methods for controlling P3HT nanowires and nanopores.

In the field of gas sensing, the preparation of nanowire structures can increase the permeability of gas, reduce the steric hindrance effect of material contact with gas, and effectively improve the detection performance of devices. The self-assembled P3HT nanofibers offer a significant advantage over earlier structures in terms of crystallinity and macroscopic charge transport. This advantage arises from the robust π–π interactions among their conjugated chains. Additionally, the fibrous aggregates of P3HT help reduce steric hindrance, resulting in enhanced gas permeability. P3HT nanowires exhibit a notable carrier mobility of around 0.01 cm^2^V^−1^s^−1^ [31], underscoring their potential application prospects in the gas sensing field. The techniques for preparing P3HT nanowires can be categorized into three approaches: (1) The dilute solution cooling method, as exemplified by Guo et al. [32], wherein P3HT was dissolved in nonideal solvents, then heated until dissolved, and subsequently left to cool for fabricating P3HT nanowires. The results demonstrated that P3HT’s crystallinity was linked to the ultrasonication time, with crystallinity increasing proportionally with extended ultrasonication; however, the nanowires prepared by this method may have a certain inconsistency in size and shape due to variations in temperature, humidity, and air pressure, which may affect the performance and repeatability of the sensor. (2) The ultrasonication method; for instance, a 1-minute ultrasonication yielded nanowires 30 nm wide and 150–200 nm long, and as ultrasonication time was extended, the quantity of nanowires grew, while their size remained relatively constant without further enlargement. When the ultrasonication time exceeded three minutes, P3HT’s crystallinity exhibited minimal alteration [33]. The P3HT nanowire sensor prepared by the ultrasonic method has some challenges in terms of stability and service life, but this method is more convenient and the prepared nanowire has good dispersion in solution. (3) The UV-induced method also explored the connection between UV irradiation and P3HT crystallization. They observed that, initially, P3HT molecular chains dispersed uniformly in the solvent, with π-electrons localized within the thiophene ring via π-orbital superposition. Under UV irradiation, P3HT molecules absorbed light, transitioning from their ground state to a photoexcited state, intensifying π-orbital superposition and enhancing the coplanarity of the molecular backbone. Consequently, P3HT molecules self-assembled into nanowire structures with prolonged exposure to UV light [34]. In the process of preparing P3HT nanowires by this method, it is difficult to ensure accurate control of the size and structure of the nanowires, especially in large-scale preparations. However, the nanowires prepared by this method are low-cost and have a high specific surface area, which can greatly improve the sensitivity of the sensor to gas.

Polymer materials with nanoporous structures possess a significantly larger specific surface area, and these porous structures create convenient channels for gas–carrier interactions. This can effectively enhance the efficiency of charge transport and is a pivotal factor in improving the gas-sensing performance of PFET sensors. There are three methods summarized for the preparation of P3HT nanopores: (1) The externally guided approach involves introducing polystyrene microspheres onto the dielectric layer’s surface and then removing them to generate a nanoporous structure. This method was used to create PFET gas sensors based on dinaphtho[2,3-b:2′,3′-f]thieno[3,2-b]thiophene (DNTT)-sensitive films for NH_3_ detection, achieving a relative sensitivity of up to 340% parts per million (ppm)^−1^ when exposed to 10 parts per billion (ppb) NH_3_ [35]. Although this method of preparing nanopores by introducing other materials can easily prepare excellent structures, it is difficult to completely remove the introduced materials from P3HT, and it may have adverse effects on the structure of P3HT nanopores. (2) The doped composite film method, as employed by Xie et al. [36], introduced a new component, reduced graphene oxide (RGO), onto a P3HT organic polymer material and prepared a gas sensor based on the nanoporous structure of a P3HT/RGO bilayer film. The stability and durability of P3HT nanopore structure prepared by this method is a key issue. For practical applications, the long-term use may lead to the change of nanopore structure, thus affecting the performance of the sensor. (3) The template-assisted method, explored by Roderick et al. [37], utilized colloidal templates in an electro-polymerization process that combined a conducting polymer with a polymer brush. Surface-initiated atom transfer radical polymerization (SI-ATRP) was then applied to cultivate the polymer, resulting in a nanoporous-like structure. Compared with previous methods, a template method can precisely control the size, shape, and distribution of nanopores. This precision can help to achieve the requirements of specific gas sensors. However, the cost is slightly higher.

At present, gas sensors using P3HT as the active layer still have some drawbacks: (1) low sensitivity and selectivity; (2) insufficient stability; and (3) complex preparation methods. Despite these disadvantages, PFETs still have potential advantages in many applications, especially in areas where flexible, lightweight, and low-cost sensors are needed. Each of the methods we have summarized above has its own advantages and limitations, conferring suitability for specific applications. Researchers must consider factors such as structural characteristics, preparation conditions, and material costs when selecting a method. No matter which method is chosen, further research and optimization are needed to improve stability, selectivity, and overall performance for successful application in the field of gas sensing.

## 3. Application of P3HT in Gas Detection

The active layer in a PFET gas sensor plays a crucial role in determining both its electrical and gas-sensitive properties. Techniques such as the micromanipulation of OSC materials, such as nanowires, and the fabrication of nanoporous materials are commonly employed to enhance the performance of these sensors.

### 3.1. P3HT Nanowire Structure in Gas Detection

Regarding the microstructure control of the P3HT polymer as an OSC, the manipulation of the nanowire structure can significantly enhance the device’s charge transfer capability. Furthermore, this manipulation can effectively regulate the carrier doping and de-doping processes [38,39,40]. Given its simplicity and practicality, this approach has gained considerable attention and has been extensively documented in recent years in gas sensors featuring P3HT as the sensing layer.

#### 3.1.1. Preparation of P3HT Nanowires by Photoinduction and Solvent Evaporation

Kim et al. [41] employed a deionized surface solvent evaporation technique to induce the self-assembly of a P3HT toluene solution into a uniform and expansive nanowire structure (in Figure 4a). This structure can be conveniently transferred to alternative substrates while maintaining its integrity. This method boasts not only simplicity and a rapid film formation rate, but it also holds potential for large-area film preparation, offering universal applicability and a high degree of reproducibility. In a related study, Nawrocki et al. [42] undertook modification of the microstructure of the P3HT film. They initiated the process by depositing a small quantity of solvent onto the pre-existing P3HT film, subsequently allowing for gradual evaporation. This technique resulted in a significant enhancement in device mobility while simultaneously preserving the initial exceptional switching ratio (in Figure 4b). Qiu et al. [43] demonstrated that by utilizing an edge solvent, they were able to attain electronic performance comparable to that of the initial P3HT material, even when employing a lower P3HT content combined with a mixture of P3HT nanowires and polystyrene (PS) (in Figure 4c). To enhance the sensing capabilities of the device, Rawlings et al. [44] introduced a universally applicable approach. This method involved utilizing an anhydrous polymer ion solution to modulate the charge accumulation mode of the semiconductor.

Moreover, Han et al. [45] employed P3HT and PS non-semiconductor layers to meticulously regulate the nanoporous morphology of the polymer through a respiration diagram. Ultimately, this manipulation resulted in an elevated current response, even in scenarios characterized by low ammonia concentrations. Noh et al. [46] utilized a straightforward wire-wound rod coating technique to attain extensive coverage, cost-effectiveness, exceptional uniformity, and precise molecular-level deposition. With an impressive accuracy level, as fine as 1.5 nm, this method yielded an PFET-based gas sensor showcasing an 82% responsiveness at an ammonia concentration of 10 ppm. In addition to this, Giridharagopal et al. [47] devised a straightforward operational approach to enhance the volume capacitance of polymer nanowires. Through their investigations, they discovered that well-structured polymer nanowires exhibited accelerated kinetics, implying a higher mobility of ions within the OSC. In an alternative approach, Sung et al. [48] made a PFET sensor for a P3HT nanowire array using photolithography technology, which realized an efficient response to ammonia gas, and not only had electrical characteristics about two orders of magnitude higher than other P3HT sensors but also had strong stability (in Figure 4d). In addition to its primary use, the nanoporous structure is also a common sensing layer microstructure of PFET-based gas sensors [49].

**Figure 4 sensors-23-08309-f004:**
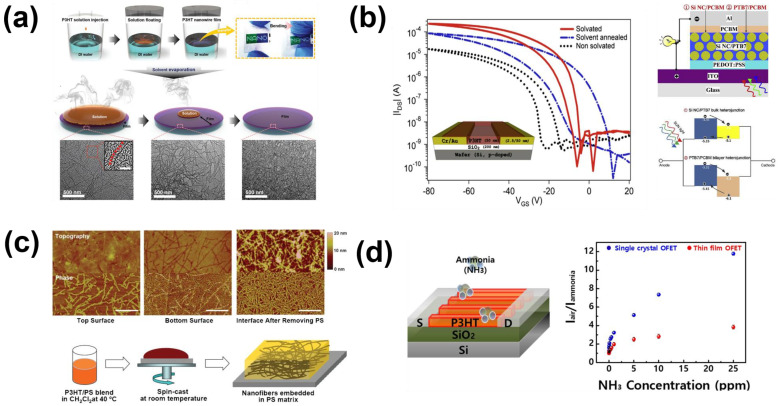
(**a**) P3HT nanowire films were synthesized using a one−pot method, wherein the microstructuring of P3HT into nanowire formations was induced through solvent evaporation on the surface of deionized water. (**b**) When the threshold voltage is 50 V, the transmission voltage curve of the three devices is changed. (**c**) Atomic force microscopy (AFM) after end−on phase separation of P3HT and schematic diagram of the production process. (**d**) Schematic diagram of NH_3_ detection by PFET nanowire structure prepared by P3HT, and performance curve. (**a**) Reproduced with permission: [41], copyright 2017, John Wiley and Sons. (**b**) Reproduced with permission: [42] copyright 2016, Elsevier. (**c**) Reproduced with permission: [43] copyright 2009, John Wiley and Sons. (**d**) Reproduced with permission: [48] copyright 2017, American Chemical Society.

#### 3.1.2. P3HT Nanowires Prepared by External Environment Control Method

In addition to the self-assembly of nanowire microstructures through spin coating, researchers have explored numerous alternative technical approaches to the cultivation of nanowire microstructures. These include: (1) manipulating microstructure morphology by capitalizing on the distinct solubility of polymers in solvents while judiciously adjusting temperature; and (2) combining polymers with other organic or inorganic materials has been investigated to yield targeted structures. These strategies have been extensively documented within the current body of research. Tang et al. [50] used an ultra-low concentration of P3HT to prepare a uniform and well-dispersed nanowire network, appropriately doped with other compounds, and the charge mobility of the optimized P3HT nanowire network increased by four orders of magnitude, which laid a certain foundation for the development of conjugated polymers devices. In a recent study, Bastianini et al. [51] employed visible spectrum technology to investigate the spatial behavior of P3HT within a chlorobenzene solution. They observed a gradual transformation of P3HT molecules from a rod-like structure into a two-dimensional lamellar nanowire configuration over time (in Figure 5a). Furthermore, in a study conducted last year, Shin et al. [52] introduced a straightforward technique involving the pre-fabrication of nanowire structures. By employing thermal annealing, they achieved a substantial enhancement in charge transport and an increased surface area ratio of the P3HT polymer film in contact with the test material (in Figure 5b). The authors conclude that this strategy holds the potential to significantly enhance the overall performance of PFET-based sensors for volatile organic compounds (VOCs). Moreover, Sadowski et al. [53] blended P3HT with phenylC61-butyric acid methyl ester (PCBM) at a specific concentration ratio to fabricate nanowires ranging from 200 to 500 nm in length (in Figure 5c). This strategic combination notably enhanced the sensing attributes of the device. Kuo et al. [54] made use of P3HT and metal oxide to prepare an organic inorganic gas sensor for NH_3_ detection. They wrapped P3HT on the outside of ZnO nanowires, and effectively retained the good high hole mobility and electrical conductivity of P3HT; the resulting device could reach a sensitivity of 11.58 ppm. Furthermore, in addition to the aforementioned approaches, Hart et al. [55] investigated localized thermal annealing and recrystallization within a roll-to-roll mode. They achieved the creation of a nanowire structure on the chloroform active layer of the P3HT polymer through printing techniques (in Figure 5d). This innovative method significantly contributed to the enhanced overall performance of the device. In a paper published in March this year, Jeong et al. [56] reported a method for NO_2_ detection by combining P3HT nanowires with nanopore structures. They combined the nanowires’ efficient charge transport pathway with the nanoporous and efficient gas-sensitive properties, resulting in a detection limit of less than 0.1 ppm and a response time of around 100 s.

### 3.2. P3HT Nanoporous Structure in Gas Detection

Because of the relatively weak interactions between gas molecules and OSCs, the signal conversion capacity of PFET-based gas sensors remains somewhat limited. In recent years, researchers have pursued a strategy of self-assembling the rigid conjugated skeleton of the polymer into nanoporous-like structures through micro-regulation. This approach is advantageous for the sensing performance of the device because the enhanced nanoporous crystallinity within the polymer chain skeleton of the active layer effectively improves carrier mobility [57,58,59,60,61,62,63,64].

#### 3.2.1. P3HT Nanoporous Structures Prepared by Photolithography and Spin Coating

Pernites et al. [37] investigated the formation of a self-assembled two-dimensional patterned surface utilizing a single layer of semiconductor polymer (in Figure 6a). Utilizing an instrument, they successfully demonstrated the creation of well-defined, elongated microstructures. However, despite this notable achievement, they did not proceed to practically implement this specific structure. In contrast to the approach employed by Pernites’ research group, Guo et al. [65] took a different route and avoided using a template. They accomplished this by blending polyethylene glycol (PEG) with PS and employing spin coating to induce phase separation and create nanopores (in Figure 6b). Through subsequent processes, they effectively achieved functionalization of the microstructure. Hulkkonen et al. [66] employed photolithography similarly, focusing on refining traditional photolithography techniques. They employed block copolymers (BCPs) to modify self-assembly, enabling the effective tailoring of the structure of OSCs without being limited by the size constraints of conventional photolithography (in Figure 6c). This approach has yielded significant breakthroughs in large-scale preparation. Wang et al. [67] built upon the work of previous researchers to optimize the device, culminating in practical testing. The sensor’s detection limit for NO_2_ in the environment reached an impressive 20 ppb, showcasing a substantial enhancement in sensitivity (in Figure 6d). Darshan et al. [68] also conducted practical experiments using the polymer devices they prepared. They successfully developed a PFET-based NH_3_ sensor, demonstrating an impressive detection limit of up to 100 ppb (in Figure 6e). Tran et al. [69] employed a nanoporous structure in their gas sensor design to detect NH_3_. They utilized the shear-assisted phase separation method to finely control the morphology of nanopores, resulting in the attainment of specific pore sizes. By manipulating the shear rate through controlling the pore size, they demonstrated exceptional selectivity for NH_3_, even within complex environments. In a similar vein, Morgera et al. [70] undertook the optimization of the surface topography of P3HT, resulting in the development of a gas sensor capable of the real-time monitoring of acetone (in Figure 6f).

#### 3.2.2. The Pore Structure of P3HT Prepared by Respiration Pattern and Doping Method

Indeed, recent years have witnessed the emergence of numerous alternative methods for crafting nanopore structures. As mentioned above [38], through the combination of P3HT and PS as the semiconductor layer, the controlled morphology of the blended film was beneficial for effective gas sensing, enabling the efficient detection of ammonia gas. This sensor demonstrated exceptional performance characteristics, including commendable detection limit, sensitivity, selectivity, and stability when applied in clinical settings. Furthermore, their work successfully achieved signal amplification specific to certain gases. Yu et al. [71] used the same blending method in which poly(methylmethacrylate) (PMMA) is blended with P3HT and optimizes the ratio (in Figure 7a). The use of this method confers the sensor surface with better nitrogen dioxide adsorption ability, which greatly improves the sensitivity and selectivity compared to previous P3HT as the sensing layer. Similar to the research conducted by Yu’s team, Chuang et al. [72], also employed a blending approach involving P3HT, this time in combination with n-type OSCs (in Figure 7b). Through this method, they successfully achieved a response to NH_3_ at the ppb level. Shalu et al. [73] enhanced the solvent composition and conducted a comprehensive examination of the impact of chlorobenzene and chloroform on the microstructure of P3HT (in Figure 7c). Their findings indicated that chlorobenzene exhibited superior response performance for the device. Subsequently, they created nanorods from a metal oxide (ZnO) that were combined with P3HT. This resultant nanoporous structure exhibited commendable device performance. Lee et al. [74] conducted an intricate investigation into the solution self-assembly mechanism of the P3HT/zinc salt complex system (in Figure 7d). Through meticulous control of the solution’s aging time, they managed to tailor a suitable micro-morphology within the solution.

Furthermore, they successfully transferred these self-assembled nanostructures to alternative substrates, effectively demonstrating their capability to manipulate and utilize these structures. Liu et al. [75] took a further step, enhancing the existing respiration diagrams through the combination of P3HT and polystyrene-block-poly(ethylene-ran-butylene)-block-polystyrene (SEBS) thin films (in Figure 7e). This innovative blending led to increased active sites and more effectively dispersed stresses during self-assembly. Their efforts yielded remarkable results, including a minimum sensitivity of 476 ppm and an impressive detection limit of 2.45 ppb. In January of the current year, Jeong et al. [76] published a recent article employing P3HT nanopore structures for gas detection. Notably, their prepared device exhibited significantly improved performance compared to the unmodified original setup (in Figure 7f). The detection limit for nitric oxide (NO) was notably reduced to 0.5 ppm, accompanied by a response time of approximately 7 min.

Considering the present landscape, the prevailing trend involves synthesizing novel compounds by optimizing and modifying existing materials or fine-tuning the side chains of materials. This approach holds significant and invaluable implications for guiding the future development and utilization of high-performance PFET-based gas sensors. We summarize some of these representative works in Table 1.

## 4. Conclusions and Future Outlook

The advancement of high-performance PFETs, driven by process engineering and meticulous micro-morphology control, has undergone substantial strides in recent years. This progress serves as the bedrock for the forthcoming generation of marker-free, cost-effective, portable, and lightweight sensors. These sensors hold tremendous potential for a multitude of outcomes in gas detection within intricate environments, yielding a substantial body of outcomes [77,78,79].

P3HT is a promising sensing material due to its stable sensing properties and mobility, which enhance its surface reactivity with analytes. In recent years, field-effect sensors based on P3HT have rapidly emerged for gas-sensing applications. Various approaches have been explored to further improve the sensitivity and specificity of P3HT materials for target gases, including: (1) P3HT nanowires; (2) P3HT nanopores. The formation of P3HT nanowires facilitates the selective penetration of target gases and minimizes the negative impact of interfering analytes. As a result, the structure of P3HT nanowires has garnered significant attention in the field of gas sensing. However, with regard to current preparation methods, certain challenges persist. For instance, P3HT nanowires prepared through dilute solution cooling and UV-induced methods, while straightforward, are highly sensitive to external environmental factors such as temperature, humidity, and pressure. This sensitivity hinders large-scale commercial manufacturing and presents ongoing challenges for further development. In addition, the preparation of P3HT nanopores can effectively enhance the material’s gas permeability and reduce the steric hindrance effect of gases. This method is also effective in improving gas sensitivity in sensors. Among the three methods summarized in this paper, both the externally guided approach and the doped composite film method tend to have adverse effects on the nanopore structure of P3HT during the preparation process. Therefore, the manufacturing process requires strict precision and is time-consuming. In comparison to the first two methods, the template method allows for the precise control over the size, shape, and distribution of the porous structure, to meet specific gas sensor requirements. Furthermore, it offers strong advantages in terms of repeatability and manufacturing consistency, although cost remains a concern [80,81,82].

After discussing various factors such as preparation methods, cost control, repeatability, and more, it is evident that using P3HT as the active layer in gas sensors presents certain challenges. However, PFET technology still holds significant potential advantages, particularly in applications demanding flexible, lightweight, and cost-effective sensors. In summary, each of the methods we have outlined comes with its own set of advantages and limitations. Researchers must carefully weigh factors such as structural characteristics, preparation conditions, material costs, and other considerations when selecting the most suitable approach.

In conclusion, this comprehensive review provides an in-depth and thorough overview of the recent advancements in the field of PFETs utilizing P3HT as the sensing layer. The intention is that this review serves as a source of inspiration for researchers in the field, offering insights that can foster further advancements and progress in the realm of gas sensing technology. The hope is that this synthesis of knowledge will contribute to the continued development and enhancement of gas sensing technologies.

## Figures and Tables

**Figure 1 sensors-23-08309-f001:**
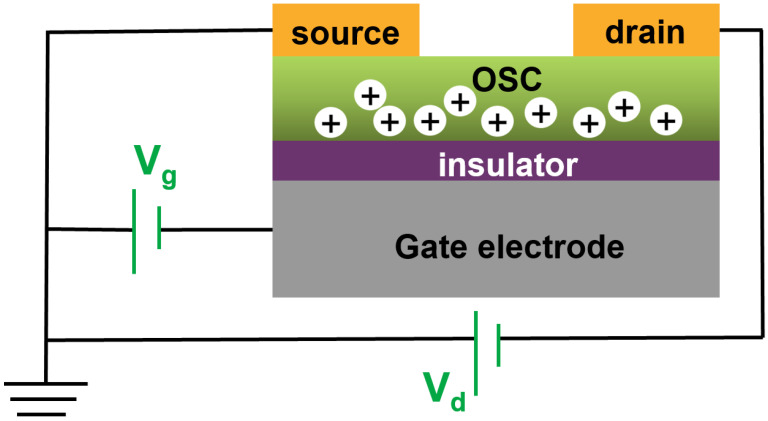
Schematic diagram of PFET structure and operating mechanism.

**Figure 2 sensors-23-08309-f002:**
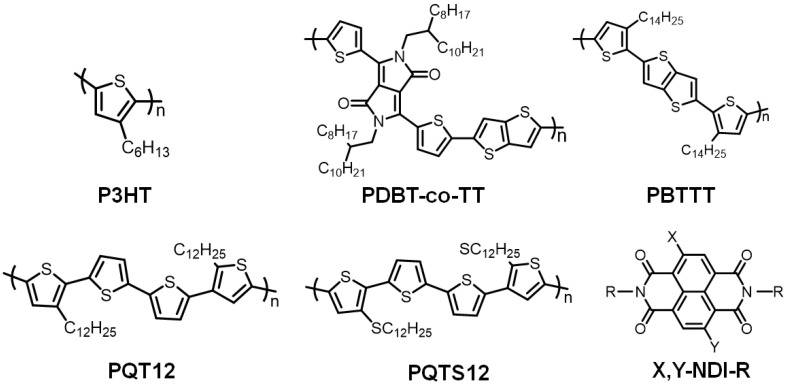
Common organic polymer OSCs materials.

**Figure 3 sensors-23-08309-f003:**
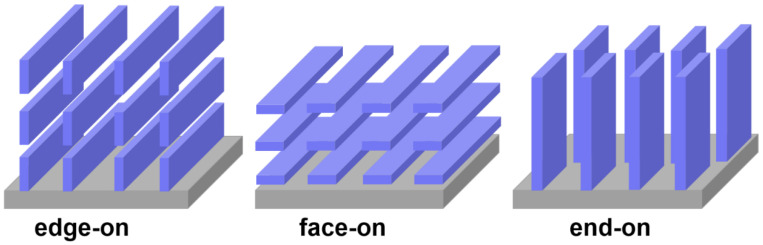
Three different stacking structures of P3HT molecular chains.

**Figure 5 sensors-23-08309-f005:**
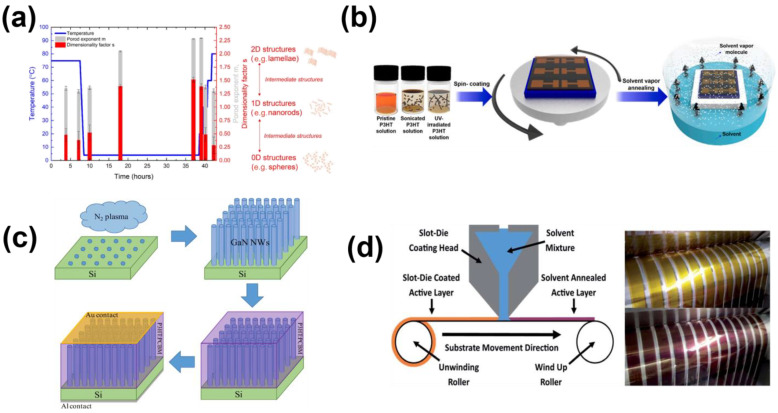
(**a**) Time and temperature curve. Temperature (blue line and scale), M-index (gray bar and red scale) and dimension factors (red bar and scale) are functions of the aging time of P3HT solution. (**b**) Diagram of the P3HT solvent vapor annealing (SVA) process. (**c**) Preparation process diagram: Preparing silicon nitride layer for nanowire nucleation. (**d**) Solvent annealing tank die coating steps diagram. The yellow is the active film printed by the slot die, and the red is the active film after solvent annealing. (**a**) Reproduced with permission: [51] copyright 2019, Elsevier. (**b**) Reproduced with permission: [52] copyright 2022, Elsevier. (**c**) Reproduced with permission: [53] copyright 2020, Springer Nature. (**d**) Reproduced with permission: [55] copyright 2019, RSC Publishing.

**Figure 6 sensors-23-08309-f006:**
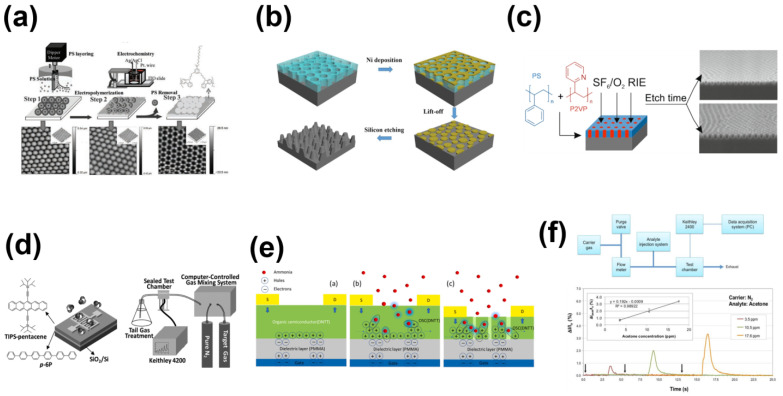
(**a**) The P3HT network monolayer array is used for mapping and enlarging 2D AFM terrain images. (**b**) Schematic diagram of nanoporous nanostructures prepared by spin coating phase separation lithography for P3HT surface modification. (**c**) Schematic diagram of a gas sensor made of a nanoporous film mask, and transmission curve. (**d**) Structure of the sensor device and molecular structure of the material and schematic diagram of the sensor test system. (**e**) Schematic diagram of analyte–semiconductor interaction. (**f**) The schematic diagram illustrates the testing principle of the device, along with the test curve depicting the response of the P3HT nanowire sensor to varying acetone concentrations at room temperature. (**a**) Reproduced with permission: [37] copyright 2011, John Wiley and Sons. (**b**) Reproduced with permission: [65] copyright 2015, Springer Nature. (**c**) Reproduced with permission: [66] copyright 2017, American Chemical Society. (**d**) Reproduced with permission: [67] copyright 2017, John Wiley and Sons. (**e**) Reproduced with permission: [68] copyright 2021, Elsevier. (**f**) Reproduced with permission: [69] copyright 2019, MDPI.

**Figure 7 sensors-23-08309-f007:**
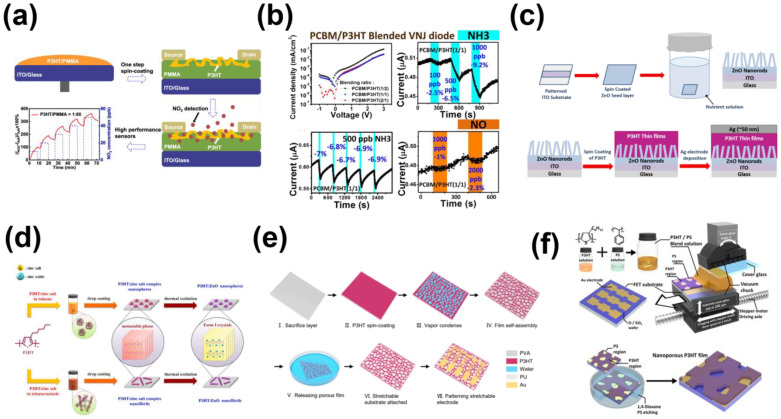
(**a**) Schematic diagram of the device spin coating process, schematic diagram of the interaction of NO_2_ with different process sensors, and real−time gas sensing curve. (**b**) Electrical characteristic diagram of P3HT blended with different materials and response curve of 5 induction cycles at fixed ammonia concentration. (**c**) Schematic diagram of preparation of ZnO nanorods by hydrothermal method and fabrication procedure of hybrid photodetector. (**d**) A schematic illustration of effective control of solution self−assembly for P3HT/zinc salt complexes showing different in situ template-synthesized nanometer P3HT/ZnO hybridization and corresponding crystal structure transformation of P3HT matrix. (**e**) Principle and preparation procedures of the breath-figure method. (**f**) Schematic diagram of preparation of P3HT/PS blended membrane and preparation of nanoporous P3HT membrane using SAPS method and selective solvent etching. (**a**) Reproduced with permission: [71] copyright 2019, American Chemical Society. (**b**) Reproduced with permission: [72] copyright 2016, Elsevier. (**c**) Reproduced with permission: [73] copyright 2019, Elsevier. (d) Reproduced with permission: [74] copyright 2021, Elsevier. (**e**) Reproduced with permission: [75] copyright 2022, Elsevier. (**f**) Reproduced with permission: [76] copyright 2023, MDPI.

**Table 1 sensors-23-08309-t001:** Comparison of the sensing performance of P3HT-based gas sensors prepared by different processes.

Materials	Analytes	Property	Device	Reference
P3HT nanowires	NH_3_	5 ppm	BGTC	[44]
P3HT nanowires	NH_3_	25 ppm	BGTC	[47]
P3HT nanowires	NH_3_	11.58 ppm	BGTC	[53]
P3HT nanowires	NO_2_	38.2%	BGTC	[55]
P3HT nanopores	NH_3_	100 ppb	BGTC	[64]
P3HT nanopores	NH_3_	70.7%	TGTC	[65]
P3HT nanopores	Acetone	3.5 pm	TGTC	[66]
P3HT nanopores	NO_2_	0.7 ppb	BGTC	[67]
P3HT nanopores	NO_2_	100 ppb	BGTC	[68]
P3HT nanopores	NH_3_	2.45 ppb	BGTC	[71]
P3HT nanopores	NO	0.5 ppm	BGTC	[72]

## Data Availability

Not applicable.

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
