# Peer review of "Recent Progress in Gas Sensors Based on P3HT Polymer Field-Effect Transistors"

_sensors, 2023, doi:10.3390/s23198309_

Round 1

Reviewer 1 Report

The authors describe in this paper an exhaustive overview about the production, functionalization strategies, and practical applications of gas sensors incorporating P3HT as the OSC layer. I consider that the article is suitable for publication. Only minor revisions are required:

1.     In section 2.1, between lines 75-90, I miss some references about the statements mentioned in the article.

2.     Between lines 134-151, please, include more references that support the statements described in the article.

3.     Line 194, there is a mistake (thio-phene). Write it correctly.

More detail:

- The article is an exhaustive overview about the production, functionalization strategies, and practical applications of gas sensors incorporating P3HT as the OSC layer

- To improve the quality of the article they should introduce more references or more research, but the article would be too long.

- The tables and figures appropriate for the article.

Author Response

The authors describe in this paper an exhaustive overview about the production, functionalization strategies, and practical applications of gas sensors incorporating P3HT as the OSC layer. I consider that the article is suitable for publication. Only minor revisions are required:

Comment 1. In section 2.1, between lines 75-90, I miss some references about the statements mentioned in the article.

Response: Thank you for the valuable suggestion. We have supplemented the corresponding reference. [18] Yin, X.; Yang, J.; Wang, H. Vertical phase separation structure for high-performance organic thin-film transistors: Mechanism, optimization strategy, and large-area fabrication toward flexible and stretchable electronics. Adv. Funct. Mater. 2022, 32, 22020.

Comment 2. Between lines 134-151, please, include more references that support the statements described in the article.

Response: Thank you for the valuable suggestion. We have supplemented the corresponding references. [27] Wu, M.; Hou, S.; Yu, X.; Yu, J. Recent progress in chemical gas sensors based on organic thin film transistors. J. Mater. Chem. C 2020, 8, 13482-13500. [28] Liu, X.; Zheng, W.; Kumar, R.; Kumar, M.; Zhang, J. Conducting polymer-based nanostructures for gas sensors. Coord. Chem. Rev. 2022, 462, 214517. [29] Yang, G. G.; Kim, D. H.; Samal, S.; Choi, J.; Roh, H.; Cunin, C. E.; Lee, H. M.; Kim, S. O.; Dincă, M.; Gumyusenge, A. Polymer-based thermally stable chemiresistive sensor for real-time monitoring of NO2 gas emission. ACS Sens. 2023. 10.1021/acssensors.3c01530. [30] Su, Y. W.; Lin, Y. C.; Wei, K. H. Evolving molecular architectures of donor-acceptor conjugated polymers for photovoltaic applications: From one-dimensional to branched to two-dimensional structures. J. Mater. Chem. A 2017, 5, 24051-24075.

Comment 3. Line 194, there is a mistake (thio-phene). Write it correctly.

Response: Thank you a lot for the positive evaluation of our manuscript. We have corrected “thio-phene” into “thiophene”.

Reviewer 2 Report

Amid the soaring global energy consumption and consequent toxic gas emissions, the polymer field effect transistor (PFET)-based gas sensor has emerged as a cost-effective solution, gaining traction for its material diversity, flexibility, and affordability. This paper delves into functional organic semiconductors, especially spotlighting poly(3-hexylthiophene-2,5-diyl) (P3HT) for its outstanding sensing capabilities and accessibility. Serving as a comprehensive guide, the study explores P3HT's production, functionalization, and application in gas sensing, underscoring its potential to enhance PFET gas sensors. The content is organized and updated. I would recommend this review to be accepted after following revision.

  1. Figure Clarity: For a review paper, figures play an integral role in conveying complex ideas in a comprehensible manner. Unfortunately, the figures in the current manuscript suffer from low resolution and appear small. To enhance readability and clarity, please consider improving the resolution and enlarging the figures.

  2. Chemical Structures in Figure 2: The chemical structures of PDBT-co-TT and NDI as depicted in Figure 2 appear to be incorrect. Please double-check the structures against reputable sources and make necessary corrections to ensure accuracy.

  3. Terminology & Citation: In the manuscript, the term "vertical" is used. For consistency and clarity within the field, it would be more appropriate to use the term "end-on." Additionally, kindly cite the following reference when making this change: J. Mater. Chem. A, 2017,5, 24051-24075.

  4. Comparative Table: One of the strong suits of a review paper is its ability to provide a comprehensive overview of a topic. The third section of the manuscript could benefit significantly from a comparative table. This would allow readers to quickly understand and compare the various topics covered, adding value to the review. Please consider incorporating such a table for a more in-depth yet concise comparison.

These comments aim to enhance the quality and clarity of the manuscript, ensuring it provides maximum value to readers in the field.

Author Response

Amid the soaring global energy consumption and consequent toxic gas emissions, the polymer field effect transistor (PFET)-based gas sensor has emerged as a cost-effective solution, gaining traction for its material diversity, flexibility, and affordability. This paper delves into functional organic semiconductors, especially spotlighting poly(3-hexylthiophene-2,5-diyl) (P3HT) for its outstanding sensing capabilities and accessibility. Serving as a comprehensive guide, the study explores P3HT's production, functionalization, and application in gas sensing, underscoring its potential to enhance PFET gas sensors. The content is organized and updated. I would recommend this review to be accepted after following revision.

Comment 1. Figure Clarity: For a review paper, figures play an integral role in conveying complex ideas in a comprehensible manner. Unfortunately, the figures in the current manuscript suffer from low resolution and appear small. To enhance readability and clarity, please consider improving the resolution and enlarging the figures.

Response: Thank you for this suggestion. We have reinserted the image and changed the resolution.

Comment 2. Chemical Structures in Figure 2: The chemical structures of PDBT-co-TT and NDI as depicted in Figure 2 appear to be incorrect. Please double-check the structures against reputable sources and make necessary corrections to ensure accuracy.

Response: Thank you. We have modified the chemical structure of PDBT-co-TT and NDI.

Comment 3. Terminology & Citation: In the manuscript, the term "vertical" is used. For consistency and clarity within the field, it would be more appropriate to use the term "end-on." Additionally, kindly cite the following reference when making this change: J. Mater. Chem. A, 2017,5, 24051-24075.

Response: Thank you for this important suggestion. We have changed “vertical” to “end-on” in the corresponding paragraph and supplemented the relevant literature. Please find on page 4-5.

Comment 4. Comparative Table: One of the strong suits of a review paper is its ability to provide a comprehensive overview of a topic. The third section of the manuscript could benefit significantly from a comparative table. This would allow readers to quickly understand and compare the various topics covered, adding value to the review. Please consider incorporating such a table for a more in-depth yet concise comparison.

Response: Thank you for this important suggestion. We have compiled a table comparing the sensing performance of P3HT-based gas sensors prepared by different processes. Please find on page 13.

Reviewer 3 Report

This paper reviews the state of the art in organic FETs for gas sensing. The review is timely and interesting, even though totally focused on a single type of polymer, namely, P3HT.

Authors review the current results and give a short outlook on what research directions should be taken to advance the state of the art.

The references chosen and the discussion gives a good overview of the results achieved so far. However, a review should also critically assess results and this is lacking in this paper. Besides describing the different findings and describing why these findings are of interest, the existing shortcomings should be clearly indicated as well. I give a few examples of what could be added to help authors in the revision of their manuscript:

1. For the different synthesis approaches, possible drawbacks in scalability, reproducibility or ease of integration onto the application substrate should be reported and discussed to help readers identify which method(s) might be more advantageous.

2. For any of the literature cited in which the performance of P3HT FETs is reported, it is necessary to add a short paragraph indicating what the shortcomings are. It might be because the study did not consider adding moisture cross-sensitivity studies, because the long-term stability was overlooked or because the material is shown to suffer from any of these problems (or others).

3. By implementing the prior 2 recommendations, the Conclusions section might need to be rewritten to clearly indicate what are the most important drawbacks that need to be addressed and, if possible, authors may add their own views on how these could be overcome.

Check for typos.

Author Response

This paper reviews the state of the art in organic FETs for gas sensing. The review is timely and interesting, even though totally focused on a single type of polymer, namely, P3HT. Authors review the current results and give a short outlook on what research directions should be taken to advance the state of the art.

Comment 1. For the different synthesis approaches, possible drawbacks in scalability, reproducibility or ease of integration onto the application substrate should be reported and discussed to help readers identify which method(s) might be more advantageous.

Response: Thank you for this important suggestion. According to different synthesis methods, we have summarized the advantages and disadvantages in the field of gas sensing. Please find on page 5-6.

Comment 2. For any of the literature cited in which the performance of P3HT FETs is reported, it is necessary to add a short paragraph indicating what the shortcomings are. It might be because the study did not consider adding moisture cross-sensitivity studies, because the long-term stability was overlooked or because the material is shown to suffer from any of these problems (or others).

Response: Thank you for this important suggestion. We have added a paragraph summarizing the current challenges of P3HT-based gas sensors. Please find on page 6.

Comment 3. By implementing the prior 2 recommendations, the Conclusions section might need to be rewritten to clearly indicate what are the most important drawbacks that need to be addressed and, if possible, authors may add their own views on how these could be overcome.

Response: Thank you for this important suggestion. We have revised the summary, clearly pointed out the defects of t P3HT material, and added our own ideas. Please find on page 13.

Reviewer 4 Report

This article systematically summarizes the research progress of PFET-type gas sensors using P3HT as the active layer. I suggest publication of this work after revision. Below are several issues that need to be solved.

1. The title of this article focuses on PFET, but the literature review is only about P3HT. Although P3HT is a typical polymer semiconductor with many advantages, it is also different from more cutting-edge D-A type semiconductors with high mobility. It is suggested that the title of the article should be revised to be more specific scope.

2. The literature review only shows the results of nanowires and nanopores, which is insufficient for a review paper. It is recommended to refer to these papers to further enrich the content of the article [1-10].

[1] Jang, D.; Park, S. Y.; Lee, H. S.; Park, Y. D. Low-Regioregularity Polythiophene for a Highly Sensitive and Stretchable Gas Sensor. ACS Appl Mater Interfaces 2023, 15 (27), 32629-32636.

[2] Cao, Z.; Huo, X.; Ma, Q.; Song, J.; Pan, Q.; Chen, L.; Lai, J.; Shan, X.; Gao, J. TFT-CN/P3HT blending active layer based two-component organic field-effect transistor for improved H2S gas detection. Sensors and Actuators B: Chemical 2023, 385.

[3] Tran, V. V.; Jeong, G.; Kim, K. S.; Kim, J.; Jung, H. R.; Park, B.; Park, J. J.; Chang, M. Facile Strategy for Modulating the Nanoporous Structure of Ultrathin pi-Conjugated Polymer Films for High-Performance Gas Sensors. ACS Sens 2022, 7 (1), 175-185.

[4] Kwon, E. H.; Kim, M.; Lee, C. Y.; Kim, M.; Park, Y. D. Metal-Organic-Framework-Decorated Carbon Nanofibers with Enhanced Gas Sensitivity When Incorporated into an Organic Semiconductor-Based Gas Sensor. ACS Appl Mater Interfaces 2022, 14 (8), 10637-10647.

[5] Boujnah, A.; Boubaker, A.; Pecqueur, S.; Lmimouni, K.; Kalboussi, A. An electronic nose using conductometric gas sensors based on P3HT doped with triflates for gas detection using computational techniques (PCA, LDA, and kNN). Journal of Materials Science: Materials in Electronics 2022, 33 (36), 27132-27146.

[6] Wu, M.; Hou, S.; Yu, X.; Yu, J. Recent progress in chemical gas sensors based on organic thin film transistors. Journal of Materials Chemistry C 2020, 8 (39), 13482-13500.

[7] Zhang, S.; Zhao, Y.; Du, X.; Chu, Y.; Zhang, S.; Huang, J. Gas Sensors Based on Nano/Microstructured Organic Field-Effect Transistors. Small 2019, 15 (12), e1805196.

[8] Surya, S. G.; Raval, H. N.; Ahmad, R.; Sonar, P.; Salama, K. N.; Rao, V. R. Organic field effect transistors (OFETs) in environmental sensing and health monitoring: A review. TrAC Trends in Analytical Chemistry 2019, 111, 27-36.

[9] Lienerth, P.; Fall, S.; Lévêque, P.; Soysal, U.; Heiser, T. Improving the selectivity to polar vapors of OFET-based sensors by using the transfer characteristics hysteresis response. Sensors and Actuators B: Chemical 2016, 225, 90-95.

[10] Zhang, C.; Chen, P.; Hu, W. Organic field-effect transistor-based gas sensors. Chem Soc Rev 2015, 44 (8), 2087-2107.

Author Response

This article systematically summarizes the research progress of PFET-type gas sensors using P3HT as the active layer. I suggest publication of this work after revision. Below are several issues that need to be solved.

Comment 1. The title of this article focuses on PFET, but the literature review is only about P3HT. Although P3HT is a typical polymer semiconductor with many advantages, it is also different from more cutting-edge D-A type semiconductors with high mobility. It is suggested that the title of the article should be revised to be more specific scope.

Response: Thank you for this important suggestion. We have changed the title into "Recent Progress in Gas Sensors Based on P3HT Polymer Field-Effect Transistors".

Comment 2. The literature review only shows the results of nanowires and nanopores, which is insufficient for a review paper. It is recommended to refer to these papers to further enrich the content of the article [1-10].

[1] Jang, D.; Park, S. Y.; Lee, H. S.; Park, Y. D. Low-Regioregularity Polythiophene for a Highly Sensitive and Stretchable Gas Sensor. ACS Appl Mater Interfaces 2023, 15 (27), 32629-32636.

[2] Cao, Z.; Huo, X.; Ma, Q.; Song, J.; Pan, Q.; Chen, L.; Lai, J.; Shan, X.; Gao, J. TFT-CN/P3HT blending active layer based two-component organic field-effect transistor for improved H2S gas detection. Sensors and Actuators B: Chemical 2023, 385.

[3] Tran, V. V.; Jeong, G.; Kim, K. S.; Kim, J.; Jung, H. R.; Park, B.; Park, J. J.; Chang, M. Facile Strategy for Modulating the Nanoporous Structure of Ultrathin pi-Conjugated Polymer Films for High-Performance Gas Sensors. ACS Sens 2022, 7 (1), 175-185.

[4] Kwon, E. H.; Kim, M.; Lee, C. Y.; Kim, M.; Park, Y. D. Metal-Organic-Framework-Decorated Carbon Nanofibers with Enhanced Gas Sensitivity When Incorporated into an Organic Semiconductor-Based Gas Sensor. ACS Appl Mater Interfaces 2022, 14 (8), 10637-10647.

[5] Boujnah, A.; Boubaker, A.; Pecqueur, S.; Lmimouni, K.; Kalboussi, A. An electronic nose using conductometric gas sensors based on P3HT doped with triflates for gas detection using computational techniques (PCA, LDA, and kNN). Journal of Materials Science: Materials in Electronics 2022, 33 (36), 27132-27146.

[6] Wu, M.; Hou, S.; Yu, X.; Yu, J. Recent progress in chemical gas sensors based on organic thin film transistors. Journal of Materials Chemistry C 2020, 8 (39), 13482-13500.

[7] Zhang, S.; Zhao, Y.; Du, X.; Chu, Y.; Zhang, S.; Huang, J. Gas Sensors Based on Nano/Microstructured Organic Field-Effect Transistors. Small 2019, 15 (12), e1805196.

[8] Surya, S. G.; Raval, H. N.; Ahmad, R.; Sonar, P.; Salama, K. N.; Rao, V. R. Organic field effect transistors (OFETs) in environmental sensing and health monitoring: A review. TrAC Trends in Analytical Chemistry 2019, 111, 27-36.

[9] Lienerth, P.; Fall, S.; Lévêque, P.; Soysal, U.; Heiser, T. Improving the selectivity to polar vapors of OFET-based sensors by using the transfer characteristics hysteresis response. Sensors and Actuators B: Chemical 2016, 225, 90-95.

[10] Zhang, C.; Chen, P.; Hu, W. Organic field-effect transistor-based gas sensors. Chem Soc Rev 2015, 44 (8), 2087-2107.

Response: Thank you for this important suggestion. We refer to these documents provided by you to further enrich the content of the review. Please find in the reference [80], [64], [69], [15], [59], [27], [11], [81], [82], [63], respectively.

Reviewer 5 Report

The work presented in this text is interesting, particularly given the current global context of heightened energy consumption and environmental challenges. It explores the practicality of gas sensors based on polymer field effect transistors (PFETs) and examines the potential of poly(3-hexylthiophene-2,5-diyl) (P3HT) as an organic semiconductor (OSC) layer. The focus on PFETs and P3HT offers a pragmatic approach to addressing these issues, with their respective advantages making them valuable components in the field of gas sensor development.

I suggest a broad review of the text and English, as some periods are quite long and complex. Breaking them down into smaller sentences can make them easier to understand. Although the text is technical, the introduction of some vocabulary variations can make it more interesting.

In the lines 38 to 42 “...In devices based on the principle of PFET, OSCs have been playing a role as a carrier transmission channel and a sensing element, which facilitates the transport of carriers through π bonds interconnected between molecules, serving as both the current-carrying transmission channel and the sensing element within PFETs, facilitate the transportation of charge carriers through the interconnected π bonds among molecules [10].”

                You cannot understand the written sentence, I suggest you rewrite it. Furthermore, the explanation regarding the conduction process through pi bonds is very vague and confusing, it would be interesting to improve it.

In the lines 90 to 92 Using p-type OSC material as an example, its working mechanism is depicted in Figure 1. When a negative voltage is applied between the source and gate electrodes, positive charge carriers are induced at the semiconductor interface near the dielectric layer due to electrostatic induction.

The authors present figure 1 which describes the PFET operating mechanism, however, the figure lacks description. Furthermore, I would suggest inserting an additional figure to this one which shows what curves characterize a transistor, in addition to informing what type of transistor would be a PFET/P3HT as this is important for us to know about the majority carriers and what their response in the presence of reducing/oxidizing gas.

From lines 101-112, the authors present a lot of information that contextualizes the text, however, without the necessary references. I suggest you insert the appropriate references.

Figure 3 could be better explained, as it is very simple and I don't see why it is in the text.

In the line 168 the author say: “ P3HT nanowires exhibit a notable carrier mobility of around 0.01 cm²V⁻¹s⁻¹ [24]”

This value is notable in relation to which parameter, it is necessary to provide something so that we can have it as a reference.

It is necessary to standardize the figures, as some are presented with letters to identify them and others are not. I would like to draw your attention to improving, in general, the resolution of all figures, especially those that are copyrighted.

I suggest a broad review of the text and English, as some periods are quite long and complex. Breaking them down into smaller sentences can make them easier to understand. Although the text is technical, the introduction of some vocabulary variations can make it more interesting.

Author Response

To Reviewer 5

The work presented in this text is interesting, particularly given the current global context of heightened energy consumption and environmental challenges. It explores the practicality of gas sensors based on polymer field effect transistors (PFETs) and examines the potential of poly(3-hexylthiophene-2,5-diyl) (P3HT) as an organic semiconductor (OSC) layer. The focus on PFETs and P3HT offers a pragmatic approach to addressing these issues, with their respective advantages making them valuable components in the field of gas sensor development.

Comment 1. I suggest a broad review of the text and English, as some periods are quite long and complex. Breaking them down into smaller sentences can make them easier to understand. Although the text is technical, the introduction of some vocabulary variations can make it more interesting.

In the lines 38 to 42 ...In devices based on the principle of PFET, OSCs have been playing a role as a carrier transmission channel and a sensing element, which facilitates the transport of carriers interconnected between molecules, serving as both the current-carrying transmission channel and the sensing element within PFETs, facilitate the transportation molecules [10].

Response: Thank you for this important suggestion. We have simplified and restated sentences in Page 1.

“In PFET-based devices, OSCs serve as carrier transmission channels and sensing elements. They enable the transport of charge carriers through interconnected π bonds between molecules, serving both as current-carrying transmission channels and sensing elements within PFETs [10].”

Comment 2. You cannot understand the written sentence, I suggest you rewrite it. Furthermore, the explanation regarding the conduction process through pi bonds is very vague and confusing, it would be interesting to improve it.

In the lines 90 to 92 Using p-type OSC material as an example, its working mechanism is depicted in Figure 1. When a negative voltage is applied between the source and gate electrodes, positive charge carriers are induced at the semiconductor interface near the dielectric layer due to electrostatic induction.

Response: Thank you for this important suggestion. We have simplified sentences in page 2-3.

“Taking p-type OSC material as an example, Figure 1 illustrates its working mechanism. When a negative voltage is applied between the source and the gate electrodes, positive charge carriers are injected from the source electrode under the control of the gate voltage, forming an accumulation layer at the interface of several molecular layers in contact between the OSCs material and the insulation layer. Subsequently, these positive charge carriers move within the conductive channel, resulting in the formation of source and drain currents, which are regulated by the source and drain voltages.”

Comment 3. The authors present figure 1 which describes the PFET operating mechanism, however, the figure lacks description. Furthermore, I would suggest inserting an additional figure to this one which shows what curves characterize a transistor, in addition to informing what type of transistor would be a PFET/P3HT as this is important for us to know about the majority carriers and what their response in the presence of reducing/oxidizing gas.

Response: Thank you for this important suggestion. There are already a few reviews to summarize this topic (Chem. Rev. 2019, 119, 3-35; Chem. Soc. Rev. 2015, 44, 2087-2107), therefore, we will not further discuss them here. In addition, we have cited the above several literatures and supplemented the sensing mechanism of P3HT with oxidizing and reducing gases. Please find on page 3.

“Given the transport characteristics of such devices, interactions between specific p-type OSCs and gas analytes (such as quenching, doping, and dipole effects) induce alterations in electrical properties like field effect mobility and threshold voltage. These changes enable instantaneous detection and response to external analyte stimulation. Specifically, the response mechanism of OSC gas sensors to oxidizing gas molecules can be understood as the "doping effect." Oxidizing gas molecules act as electron acceptors, resulting in a similar hole-doping effect when they interact with p-type semiconductors. This interaction provides additional holes for deep trap states or passivates hole traps within the organic semiconductor. As a result, there is an increase in conductivity and a positive drift in the surge voltage. Conversely, when OSC interacts with a reducing gas, the lone pair electrons in the reducing gas engage with OSC, leading to hole reduction when acting on p-type PFETs. Simultaneously, the reducing gas molecules infiltrate the interface to capture hole charges, reducing hole accumulation and transmission. This results in decreased carrier mobility, thus reducing the source-drain current, and a negative offset in the surge voltage [19, 20]”

Comment 4. From lines 101-112, the authors present a lot of information that contextualizes the text, however, without the necessary references. I suggest you insert the appropriate references.

Response: Thank you for the valuable suggestion. We have supplemented the corresponding references.

[21] Tao, J.; Liu, D.; Jing, J.; Dong, H.; Liu, L.; Xu, B.; Tian, W. Organic single crystals with high photoluminescence quantum yields close to 100% and high mobility for optoelectronic devices. Adv. Mater. 2021, 33, 2105466.

[22] Zhang, Q.; Lei, L.; Zhu, S. Gas-responsive polymers. ACS Macro Lett. 2017, 6, 515-522.

[23] Gong, Z.; Wang, Y.; Yan, Q. Polymeric partners breathe together: Using gas to direct polymer self-assembly via gas-bridging chemistry. Sci. China: Chem. 2022, 65, 1401-1410.

Comment 5. Figure 3 could be better explained, as it is very simple and I don't see why it is in the text.

Response: Thank you for the valuable suggestion. This paper introduces various arrangements of P3HT molecular chains, as these configurations have a direct impact on their performance and characteristics in organic electronic devices and material applications. The inclusion of Figure 3 provides a more vivid understanding, prediction, and optimization of the properties of this organic semiconductor material. This, in turn, can advance research and applications in related fields.

Round 2

Reviewer 3 Report

After the revision, I feel satisfied with the amended version of paper. It can now be accepted for publication in Sensors.

Reviewer 5 Report

All suggestions/questions were answered by the authors.